# CROSS-DOMAIN FEW-SHOT CLASSIFICATION VIA LEARNED FEATURE-WISE TRANSFORMATION

**Hung-Yu Tseng**
University of California, Merced
htseng6@ucmerced.edu

**Hsin-Ying Lee**
University of California, Merced
hlee246@ucmerced.edu

**Jia-Bin Huang**
Virginia Tech
jbhuang@vt.edu

**Ming-Hsuan Yang**
University of California, Merced
Google Research
Yonsei University
mhyang@ucmerced.edu

## ABSTRACT

Few-shot classification aims to recognize novel categories with only few labeled images in each class. Existing metric-based few-shot classification algorithms predict categories by comparing the feature embeddings of query images with those from a few labeled images (support examples) using a learned metric function. While promising performance has been demonstrated, these methods often fail to generalize to unseen domains due to large discrepancy of the feature distribution across domains. In this work, we address the problem of few-shot classification under domain shifts for metric-based methods. Our core idea is to use feature-wise transformation layers for augmenting the image features using affine transforms to simulate various feature distributions under different domains in the training stage. To capture variations of the feature distributions under different domains, we further apply a learning-to-learn approach to search for the hyper-parameters of the feature-wise transformation layers. We conduct extensive experiments and ablation studies under the domain generalization setting using five few-shot classification datasets: mini-ImageNet, CUB, Cars, Places, and Plantae. Experimental results demonstrate that the proposed feature-wise transformation layer is applicable to various metric-based models, and provides consistent improvements on the few-shot classification performance under domain shift.

## 1 INTRODUCTION

Few-shot classification (Lake et al., 2015) aims to recognize instances from *novel* categories (query instances) with only few labeled examples in each class (support examples). Among various recent approaches for addressing the few-shot classification problem, metric-based meta-learning methods (Garcia & Bruna, 2018; Sung et al., 2018; Vinyals et al., 2016; Snell et al., 2017; Oreshkin et al., 2018) have received considerable attention due to their simplicity and effectiveness. In general, metric-based few-shot classification methods make the prediction based on the similarity between the query image and support examples. As illustrated in Figure 1, metric-based approaches consist of 1) a feature encoder and 2) a metric function. Given an input *task* consisting of few labeled images (the support set) and unlabeled images (the query set) from novel classes, the encoder first extracts the image features. The metric function then takes the features of both the labeled and unlabeled images as input and predicts the category of the query images. Despite the success of recognizing novel classes sampled from *the same* domain as in the training stage (e.g., , both training and testing are on mini-ImageNet classes), Chen et al. (Chen et al., 2019a) recently raise the issue that existing metric-based approaches often do not generalize well to categories from *different* domains. The generalization ability to unseen domains, however, is of critical importance due to

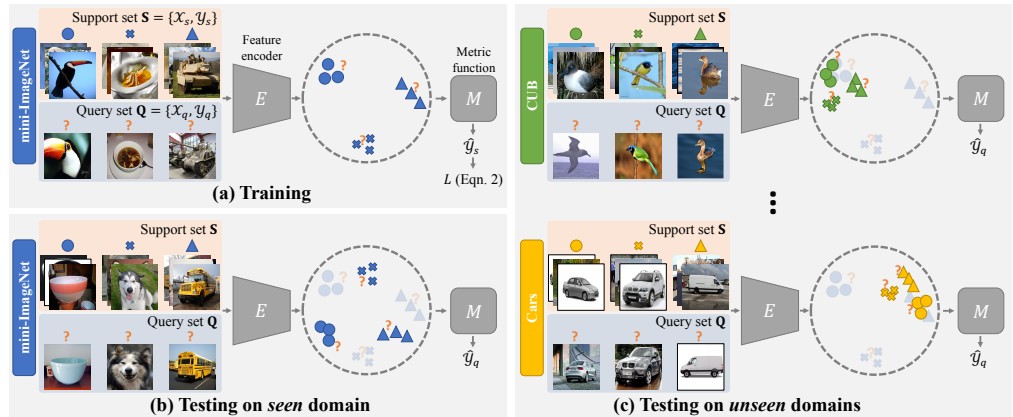

Figure 1: **Problem formulation and motivation.** Metric-based meta-learning models usually consist of a feature encoder $E$ and metric function $M$. We aim to improve the generalization ability of the models training from seen domains to arbitrary unseen domains. The key observation is that the distributions of the image features extracted from tasks in the unseen domains are significantly different from those in the seen domains.

the difficulty to construct large training datasets for rare classes (e.g., , recognizing rare bird species in a fine-grained classification setting). As a result, understanding and addressing the domain shift problem for few-shot classification is of great interest.

To alleviate the domain shift issue, numerous unsupervised domain adaptation techniques have been proposed (Pan & Yang, 2010; Chen et al., 2018; Tzeng et al., 2017). These methods focus on adapting the classifier of *the same* category from the source to the target domain. Building upon the domain adaptation formulation, Dong and Xing (Dong & Xing, 2018) relax the constraint and transfer knowledge across domains for recognizing *novel* category in the one-shot setting. However, unsupervised domain adaptation approaches assume that numerous unlabeled images are available in the target domain during training. In many cases, this assumption may not be realistic. For example, the cost and efforts of collecting numerous images of rare bird species can be prohibitively high. On the other hand, domain generalization methods have been developed (Blanchard et al., 2011; Li et al., 2019) to learn classifiers that generalize well to multiple unseen domains without requiring the access to data from those domains. Yet, existing domain generalization approaches aim at recognizing instance from *the same* category in the training stage.

In this paper, we tackle the domain generalization problem for recognizing *novel* category in the few-shot classification setting. As shown in Figure 1(c), our key observation is that the distributions of the image features extracted from the tasks in different domains are significantly different. As a result, during the training stage, the metric function may overfit to the feature distributions encoded only from the seen domains and thus fail to generalize to unseen domains. To address the issue, we propose to integrate *feature-wise transformation layer* to modulate the feature activations with affine transformations into the feature encoder. The use of these feature-wise transformation layers allows us to simulate various distributions of image features during the training stage, and thus improve the generalization ability of the metric function in the testing phase. Nevertheless, the hyper-parameters of the feature-wise transformation layers may require meticulous hand-tuning due to the difficulty to model the complex variation of the image feature distributions across various domains. In light of this, we develop a *learning-to-learn* algorithm to optimize the proposed feature-wise transformation layers. The core idea is to optimize the feature-wise transformation layers so that the model can work well on the unseen domains after training the model using the seen domains. We make the source code and datasets public available to simulate future research in this field.[1]

We make the following three contributions in this work:

- We propose to use feature-wise transformation layers to simulate various image feature distributions extracted from the tasks in different domains. Our feature-wise transformation

---

[1] https://github.com/hytseng0509/CrossDomainFewShot

layers are *method-agnostic* and can be applied to various metric-based few-shot classification approaches for improving their generalization to unseen domains.

- We develop a learning-to-learn method to optimize the hyper-parameters of the feature-wise transformation layers. In contrast to the exhaustive parameter hand-tuning process, the proposed learning-to-learn algorithm is capable of finding the hyper-parameters for the feature-wise transformation layers to capture the variation of image feature distribution across various domains.

- We evaluate the performance of three metric-based few-shot classification models (including MatchingNet (Vinyals et al., 2016), RelationNet (Sung et al., 2018), and Graph Neural Networks (Garcia & Bruna, 2018)) with extensive experiments under the domain generalization setting. We show that the proposed feature-wise transformation layers can effectively improve the generalization ability of metric-based models to unseen domains. We also demonstrate further performance improvement with our learning-to-learn scheme for learning the feature-wise transformation layers.

## 2 RELATED WORK

**Few-shot classification.** Few-shot classification aims to learn to recognize novel categories with a limited number of labeled examples in each class. Significant progress has been made using the meta-learning based formulation. There are three main classes of meta-learning approaches for addressing the few-shot classification problem. First, recurrent-based frameworks (Rezende et al., 2016; Santoro et al., 2016) sequentially process and encode the few labeled images of novel categories. Second, optimization-based schemes (Finn et al., 2017; Rusu et al., 2019; Tseng et al., 2019; Vuorio et al., 2019) learn to fine-tune the model with the few example images by integrating the fine-tuning process in the meta-training stage. Third, metric-based methods (Koch et al., 2015; Vinyals et al., 2016; Snell et al., 2017; Oreshkin et al., 2018; Sung et al., 2018; Lifchitz et al., 2019) classify the query images by computing the similarity between the query image and few labeled images of novel categories.

Among these three classes, metric-based methods have attracted considerable attention due to their simplicity and effectiveness. Metric-based few-shot classification approaches consist of 1) a *feature encoder* to extract features from both the labeled and unlabeled images and 2) a *metric function* that takes image features as input and predict the category of unlabeled images. For example, MatchingNet (Vinyals et al., 2016) applies cosine similarity along with a recurrent network, ProtoNet (Snell et al., 2017) utilizes euclidean distance, RelationNet (Sung et al., 2018) uses CNN modules, GNN (Garcia & Bruna, 2018) employs graph convolution modules as the metric functions. However, these metric functions may fail to generalize to unseen domains since the distributions of the image features extracted from the task in various domains can be drastically different. Chen et al. (Chen et al., 2019a) recently show that the performance of existing few-shot classification methods degrades significantly under domain shifts. Our work focuses on improving the generalization ability of metric-based few-shot classification models to unseen domains. Very recently, Triantafillou et al. (Triantafillou et al., 2020) also target on the cross-domain few-shot classification problem. We encourage the readers to review for a more complete picture.

**Domain adaptation.** Domain adaptation methods (Pan & Yang, 2010) aim to reduce the domain shift between the source and target domains. Since the emergence of domain adversarial neural networks (DANN) (Ganin et al., 2016), numerous frameworks have been proposed to apply adversarial training to align the source and target distributions on the feature-level (Tzeng et al., 2017; Chen et al., 2018; Hsu et al., 2020) or on the pixel-level (Tsai et al., 2018; Hoffman et al., 2018; Bousmalis et al., 2017; Chen et al., 2019b; Lee et al., 2018). Most domain frameworks, however, target at adapting knowledge of the *same* category learned from the source to target domain and thus are less effective to handle *novel* category as in the few-shot classification scenarios. One exception is the work by Dong and Xing (Dong & Xing, 2018) that address the domain shift issue in the one-shot learning setting. Nevertheless, these domain adaptation methods require access to the unlabeled images in the target domain during the training. Such an assumption may not be feasible in many applications due to the difficulty of collecting abundant examples of rare categories (e.g., rare bird species).

**Domain generalization.** In contrast to the domain adaptation frameworks, domain generalization (Blanchard et al., 2011) methods aim at generalizing from a set of seen domains to the unseen domain *without* accessing instances from the unseen domain during the training stage. Before the emerging of learning-to-learn (i.e., meta-learning) (Ravi & Larochelle, 2017; Finn et al., 2017) approaches, several methods have been proposed for tackling the domain generalization problem. Examples include extracting domain-invariant features from various seen domains (Blanchard et al., 2011; Li et al., 2018b; Muandet et al., 2013), improving the classifiers by fusing classifiers learned from seen domains (Niu et al., 2015a;b), and decomposing the classifiers into domain-specific and domain-invariant components (Khosla et al., 2012; Li et al., 2017a). Another stream of work learns to augment the input data with adversarial learning (Shankar et al., 2018; Volpi et al., 2018). Most recently, a number of methods apply the learning-to-learn strategy to simulate the generalization process in the training stage (Balaji et al., 2018; Li et al., 2018a; 2019). Our method adopts a similar approach to train the proposed feature-wise transformation layers. The application context, however, differs from prior work as we focus on recognizing *novel* category from *unseen* domains in few-shot classification. The goal of this work is to make few-shot classification algorithms robust to domain shifts.

**Learning-based data augmentation.** Data augmentation methods are designed to increase the diversity of data for the training process. Unlike the hand-crafted approaches such as horizontal flipping and random cropping, several recent approaches have been proposed to *learn* the data augmentation (Cubuk et al., 2019; DeVries & Taylor, 2017a; Lemley et al., 2017; Perez & Wang, 2017; Sixt et al., 2018; Tran et al., 2017). For instance, the SmartAugmentation (Lemley et al., 2017) scheme trains a network that combines multiple images from the same category. The Bayesian DA (Tran et al., 2017) method augments the data according to the distribution learned from the training set, and the RenderGAN (Sixt et al., 2018) model simulates realistic images using generative adversarial networks. In addition, the AutoAugment (Cubuk et al., 2019) algorithm learns the augmentation via reinforcement learning. Two recent frameworks (Shankar et al., 2018; Volpi et al., 2018) target at augmenting the data by modeling to the variation across different domains with adversarial learning. Similar to these approaches for capturing the variations across multiple domains, we develop a learning-to-learn process to optimize the proposed feature-wise transformation layers for simulating various distributions of image features encoded from different domains.

**Conditional normalization.** Conditional normalization aims to modulate the activation via a learned affine transformation conditioned on external data (e.g., an image of an artwork for capturing a specific style). Conditional normalization methods, including Conditional Batch Normalization (Dumoulin et al., 2017), Adaptive Instance Normalization (Huang & Belongie, 2017), and SPADE (Park et al., 2019), are widely used in the style transfer and image synthesis tasks (Karras et al., 2019; Lee et al., 2020; AlBahar & Huang, 2019). In addition to image stylization and generation, conditional normalization has also been applied to align different data distributions for domain adaptation (Cariucci et al., 2017; Li et al., 2017b). In particular, the TADAM method (Oreshkin et al., 2018) applies conditional batch normalization to metric-based models for the few-shot classification task. The TADAM method aims to model the training task distribution under the *same* domain. In contrast, we focus on simulating various features distributions from *different* domains.

**Regularization for neural networks.** Adding some form of randomness in the training stage is an effective way to improve generalization (Srivastava et al., 2014; Wan et al., 2013; Larsson et al., 2017; DeVries & Taylor, 2017b; Zhang et al., 2018; Ghiasi et al., 2018). The proposed feature-wise transformation layer for modulating the feature activations of intermediate layers (by applying random affine transformations) can also be viewed as a way to regularize network training.

## 3 METHODOLOGY

### 3.1 PRELIMINARIES

**Few-shot classification and metric-based method.** The few-shot classification problem is typically characterized as $N_w$ way (number of categories) and $N_s$ shot (number of labeled examples for each category). Figure 1 shows an example of how the metric-based frameworks operate in the 3-way 3-shot few shot classification task. A metric-based algorithm generally contains a feature

encoder $E$ and a metric function $M$. For each iteration during the training stage, the algorithm randomly samples $N_w$ categories and constructs a task $T$. We denote the collection of input images as $\mathcal{X} = \{\mathbf{x}_1, \mathbf{x}_2, \cdots, \mathbf{x}_n\}$ and the corresponding categorical labels as and $\mathcal{Y} = \{y_1, y_2, \cdots, y_n\}$. A task $T$ consists of a *support set* $\mathbf{S} = \{(\mathcal{X}_s, \mathcal{Y}_s)\}$ and a *query set* $\mathbf{Q} = \{(\mathcal{X}_q, \mathcal{Y}_q)\}$. The support set $\mathbf{S}$ and query set $\mathbf{Q}$ are respectively formed by randomly selecting $N_s$ and $N_q$ samples for each of the $N_w$ categories.

The feature encoder $E$ first extracts the features from both the support and query images. The metric function $M$ then predicts the categories of the query images $\mathcal{X}_q$ according to the label of support images $\mathcal{Y}_s$, the encoded query image $E(\mathbf{x}^q)$, and the encoded support images $E(\mathcal{X}_s)$. The process can be formulated as

$$\hat{\mathcal{Y}}_q = M(\mathcal{Y}_s, E(\mathcal{X}_s), E(\mathcal{X}_q)). \tag{1}$$

Finally, the training objective of a metric-based framework is the classification loss of the images in the query set,

$$L = L_{\text{cls}}(\mathcal{Y}_q, \hat{\mathcal{Y}}_q). \tag{2}$$

The main difference between various metric-based algorithms lies in the design choice for the metric function $M$. For instance, the MatchingNet (Vinyals et al., 2016) method utilizes long-short-term memories (LSTM), the RelationNet (Sung et al., 2018) model applies convolutional neural networks (CNN), and the GNN (Garcia & Bruna, 2018) scheme uses graph convolutional networks.

**Problem setting.** In this work, we address the few-shot classification problem under the domain generalization setting. We denote a domain consisting of a collection of few-shot classification tasks as $\mathcal{T} = \{T_1, T_2, \cdots, T_n\}$. We assume $N$ seen domains $\{\mathcal{T}_1^{\text{seen}}, \mathcal{T}_2^{\text{seen}}, \cdots, \mathcal{T}_N^{\text{seen}}\}$ available in the training phase. The goal is to learn a metric-based few-show classification model using the seen domains, such that the model can generalize well to an unseen domain $\mathcal{T}^{\text{unseen}}$. For example, one can train the model with the mini-ImageNet (Ravi & Larochelle, 2017) dataset as well as some public available fine-grained few-shot classification domains, e.g., CUB (Welinder et al., 2010), and then evaluate the generalization ability of the model on an unseen plants domain. Note that our problem formulation does *not* access images in the unseen domain at the training stage.

## 3.2 FEATURE-WISE TRANSFORMATION LAYER

Our focus in this work is to improve the generalization ability of metric-based few-shot classification models to arbitrary unseen domains. As shown in Figure 1, due to the discrepancy between the feature distributions extracted from the task in the seen and unseen domains, the metric function $M$ may overfit to the seen domains and fail to generalize to the unseen domains. To address the problem, we propose to integrate a feature-wise transformation to augment the intermediate feature activations with affine transformations into the feature encoder $E$. Intuitively, the feature encoder $E$ integrated with the feature-wise transformation layers can produce more diverse feature distributions which improve the generalization ability of the metric function $M$. As shown in Figure 2(b), we insert the feature-wise transformation layer after the batch normalization layer in the feature encoder $E$. The hyper-parameters $\theta_\gamma \in R^{C \times 1 \times 1}$ and $\theta_\beta \in R^{C \times 1 \times 1}$ indicate the standard deviations of the Gaussian distributions for sampling the affine transformation parameters. Given an intermediate feature activation map $\mathbf{z}$ in the feature encoder with the dimension of $C \times H \times W$, we first sample the scaling term $\gamma$ and bias term $\beta$ from Gaussian distributions,

$$\gamma \sim N(\mathbf{1}, \text{softplus}(\theta_\gamma)) \quad \beta \sim N(\mathbf{0}, \text{softplus}(\theta_\beta)). \tag{3}$$

We then compute the modulated activation $\hat{\mathbf{z}}$ as

$$\hat{z}_{c,h,w} = \gamma_c \times z_{c,h,w} + \beta_c, \tag{4}$$

where $\hat{z}_{c,h,w} \in \hat{\mathbf{z}}$ and $z_{c,h,w} \in \mathbf{z}$. In practice, we insert the feature-wise transformation layers to the feature encoder $E$ at multiple levels.

## 3.3 LEARNING THE FEATURE-WISE TRANSFORMATION LAYERS

While we can empirically determine hyper-parameters $\theta_f = \{\theta_\gamma, \theta_\beta\}$ of the feature-wise transformation layer, it remains challenging to hand-tune a generic set of parameters which are effective

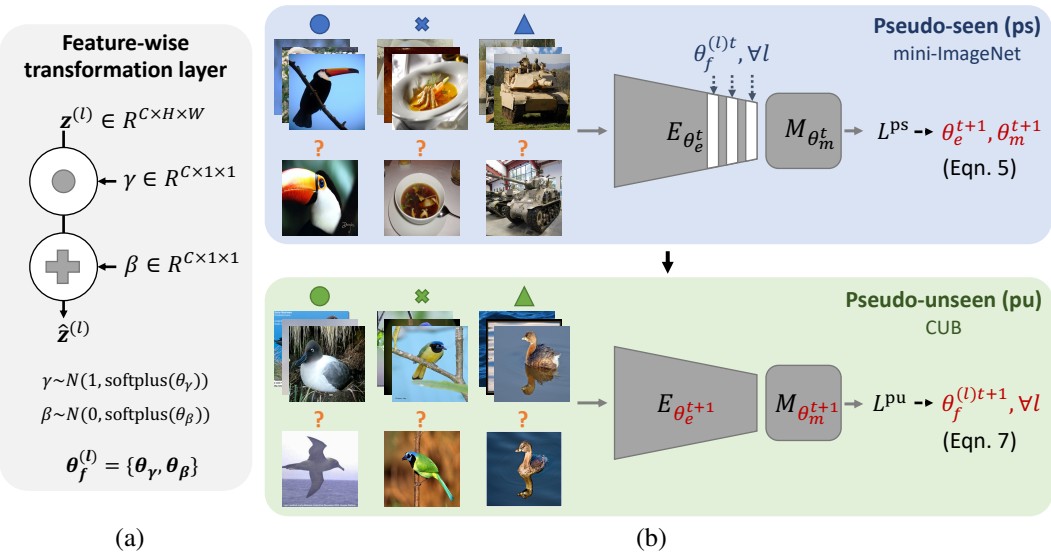

Figure 2: **Method overview.** (a) We propose a feature-wise transformation layer to modulate intermediate feature activation $\mathbf{z}$ in the feature encoder $E$ with the scaling and bias terms sampled from the Gaussian distributions parameterized by the hyper-parameters $\theta_\gamma$ and $\theta_\beta$. During the training phase, we insert a collection of feature-wise transformation layers into the feature encoder to simulate feature distributions extracted from the tasks in various domains. (b) We design a learning-to-learn algorithm to optimize the hyper-parameters $\theta_\gamma$ and $\theta_\beta$ of feature-wise transformation layers by maximizing the performance of the applied metric-based model on the pseudo-unseen domain (*bottom*) after it is optimized on the pseudo-seen domain (*top*).

on different settings (i.e., different metric-based frameworks and different seen domains). To address this problem, we design a learning-to-learn algorithm to optimize the hyper-parameters of the feature-wise transformation layer. The core idea is that training the metric-based model integrated with the proposed layers on the seen domains should improve the performance of the model on the unseen domains.

We illustrate the process in Figure 2(b) and Algorithm 1. In each training iteration $t$, we sample a *pseudo-seen* domain $\mathcal{T}^{\mathrm{ps}}$ and a *pseudo-unseen* domain $\mathcal{T}^{\mathrm{pu}}$ from a set of seen domains $\{\mathcal{T}_1^{\mathrm{seen}}, \mathcal{T}_2^{\mathrm{seen}}, \cdots, \mathcal{T}_N^{\mathrm{seen}}\}$. Given a metric-based model with feature encoder $E_{\theta_e^t}$ and metric function $M_{\theta_m^t}$, we first integrate the proposed layers with hyper-parameters $\theta_f^t = \{\theta_\gamma^t, \theta_\beta^t\}$ into the feature encoder (i.e., $E_{\theta_e^t, \theta_f^t}$). We then use the loss in equation 2 to update the parameters in the metric-based model with the *pseudo-seen* task $T^{\mathrm{ps}} = \{(\mathcal{X}_s^{\mathrm{ps}}, \mathcal{Y}_s^{\mathrm{ps}}), (\mathcal{X}_q^{\mathrm{ps}}, \mathcal{Y}_q^{\mathrm{ps}})\} \in \mathcal{T}^{\mathrm{ps}}$, namely

$$(\theta_e^{t+1}, \theta_m^{t+1}) = (\theta_e^t, \theta_m^t) - \alpha \bigtriangledown_{\theta_e^t, \theta_m^t} L_{\mathrm{cls}}(\mathcal{Y}_q^{\mathrm{ps}}, M_{\theta_m^t}(\mathcal{Y}_s^{\mathrm{ps}}, E_{\theta_e^t, \theta_f^t}(\mathcal{X}_s^{\mathrm{ps}}), E_{\theta_e^t, \theta_f^t}(\mathcal{X}_q^{\mathrm{ps}}))), \quad (5)$$

where $\alpha$ is the learning rate. We then measure the generalization ability of the updated metric-based model by 1) removing the feature-wise transformation layers from the model and 2) computing the classification loss of the updated model on the *pseudo-unseen* task $T^{\mathrm{pu}} = \{(\mathcal{X}_s^{\mathrm{pu}}, \mathcal{Y}_s^{\mathrm{pu}}), (\mathcal{X}_q^{\mathrm{pu}}, \mathcal{Y}_q^{\mathrm{pu}})\} \in \mathcal{T}^{\mathrm{pu}}$, namely

$$L^{\mathrm{pu}} = L_{\mathrm{cls}}(\mathcal{Y}_q^{\mathrm{pu}}, M_{\theta_m^{t+1}}(\mathcal{Y}_s^{\mathrm{pu}}, E_{\theta_e^{t+1}}(\mathcal{X}_s^{\mathrm{pu}}), E_{\theta_e^{t+1}}(\mathcal{X}_q^{\mathrm{pu}}))). \quad (6)$$

Finally, as the loss $L^{\mathrm{pu}}$ reflects the effectiveness of the feature-wise transformation layers, we optimize the hyper-parameters $\theta_f$ by

$$\theta_f^{t+1} = \theta_f^t - \alpha \bigtriangledown_{\theta_f^t} L^{\mathrm{pu}}. \quad (7)$$

Note that the metric-based model and feature-wise transformation layers are jointly optimized in the training stage.

---

**Algorithm 1:** Learning-to-Learn Feature-Wise Transformation.

---

1  **Require:** Seen domains $\{\mathcal{T}_1^{\text{seen}}, \mathcal{T}_2^{\text{seen}}, \cdots, \mathcal{T}_n^{\text{seen}}\}$, learning rate $\alpha$
2  Randomly initialize $\theta_e$, $\theta_m$ and $\theta_f$
3  **while** *training* **do**
4  $\quad$ Randomly sample non-overlapping pseudo-seen $\mathcal{T}^{\text{ps}}$ and psuedo-unseen $\mathcal{T}^{\text{pu}}$ domains from
$\quad\quad$ the seen domains
5  $\quad$ Sample a pesudo-seen task $T^{\text{ps}} \in \mathcal{T}^{\text{ps}}$ and a pseudo-unseen task $T^{\text{pu}} \in \mathcal{T}^{\text{pu}}$

6  $\quad$ // **Update metric-based model with pseudo-seen task**:
7  $\quad$ Obtain $\theta_e^{t+1}$, $\theta_m^{t+1}$ using equation 5

8  $\quad$ // **Update feature-wise transformation layers with pseudo-unseen task**:
9  $\quad$ Obtain $\theta_f^{t+1}$ using equation 6 and equation 7

10 **end**

---

## 4 EXPERIMENTAL RESULTS

### 4.1 EXPERIMENTAL SETUPS

We validate the efficacy of the proposed feature-wise transformation layer with three existing metric-based algorithms (Vinyals et al., 2016; Sung et al., 2018; Garcia & Bruna, 2018) under two experimental settings.First, we empirically determine the hyper-parameters $\theta_f = \{\theta_\gamma, \theta_\beta\}$ of the feature-wise transformation layers and analyze the impact of the feature-wise transformation layers. We train the few-shot classification model on the mini-ImageNet (Bousmalis et al., 2017) domain and evaluate the trained model on four different domains: CUB (Welinder et al., 2010), Cars (Krause et al., 2013), Places (Zhou et al., 2017), and Plantae (Van Horn et al., 2018). Second, we demonstrate the importance of the proposed learning-to-learn scheme for optimizing the hyper-parameters of feature-wise transformation layers. We adopt the leave-one-out setting by selecting an unseen domain from CUB, Cars, Places, and Plantae domains. The mini-ImageNet (Bousmalis et al., 2017) and the remaining domains then serve as the seen domains for training both the metric-based model and feature-wise transformation layers using Algorithm 1. After the training, we evaluate the trained model on the selected unseen domain.

**Datasets.** We conduct experiments using five datasets: mini-ImageNet (Ravi & Larochelle, 2017), CUB (Welinder et al., 2010), Cars (Krause et al., 2013), Places (Zhou et al., 2017), and Plantae (Van Horn et al., 2018). Since the mini-ImageNet dataset serves as the seen domain for all experiments, we select the training iterations with the best accuracy on the validation set of the mini-ImageNet dataset for evaluation. More details of dataset processing are presented in Appendix A.1.

**Implementation details.** We apply the feature-wise transformation layers to three metric-based frameworks: MatchingNet (Vinyals et al., 2016), RelationNet (Sung et al., 2018), and GNN (Garcia & Bruna, 2018). We use the public implementation from Chen et al. (Chen et al., 2019a) to train both the MatchingNet and RelationNet model.[2] For the GNN approach, we integrate the official implementation for graph convolutional network into Chen's implementation.[3] In all experiments, we adopt the ResNet-10 (He et al., 2016) model as the backbone network for our feature encoder $E$.

We present the average results over $1,000$ trials for all the experiments. In each trial, we randomly sample $N_w$ categories (e.g., 5 classes for 5-way classification). For each category, we randomly select $N_s$ images (e.g., 1-shot or 5-shot) for the support set $\mathcal{X}_s$ and 16 images for the query set $\mathcal{X}_q$. We discuss the implementation details in Appendix A.2.

**Pre-trained feature encoder.** Prior to the few-shot classification training stage, we first pre-train the feature encoder $E$ by minimizing the standard cross-entropy classification loss on the 64 training categories in the mini-ImageNet dataset. This strategy can significantly improve the performance of

---

[2]https://github.com/wyharveychen/CloserLookFewShot
[3]https://github.com/vgsatorras/few-shot-gnn

Table 1: **Few-shot classification results trained with the mini-ImageNet dataset.** We train the model on the mini-ImageNet domain and evaluate the trained model on another domain. FT indicates that we apply the feature-wise transformation layers with empirically determined hyper-parameters to train the model.

| 5-way 1-Shot | FT | mini-ImageNet | CUB | Cars | Places | Plantae |
|---|---|---|---|---|---|---|
| MatchingNet | - | $59.10 \pm 0.64\%$ | $35.89 \pm 0.51\%$ | $\mathbf{30.77 \pm 0.47}\%$ | $49.86 \pm 0.79\%$ | $32.70 \pm 0.60\%$ |
| | ✓ | $58.76 \pm 0.61\%$ | $\mathbf{36.61 \pm 0.53}\%$ | $29.82 \pm 0.44\%$ | $\mathbf{51.07 \pm 0.68}\%$ | $\mathbf{34.48 \pm 0.50}\%$ |
| RelationNet | - | $57.80 \pm 0.88\%$ | $42.44 \pm 0.77\%$ | $29.11 \pm 0.60\%$ | $48.64 \pm 0.85\%$ | $33.17 \pm 0.64\%$ |
| | ✓ | $58.64 \pm 0.85\%$ | $\mathbf{44.07 \pm 0.77}\%$ | $28.63 \pm 0.59\%$ | $\mathbf{50.68 \pm 0.87}\%$ | $33.14 \pm 0.62\%$ |
| GNN | - | $60.77 \pm 0.75\%$ | $45.69 \pm 0.68\%$ | $31.79 \pm 0.51\%$ | $53.10 \pm 0.80\%$ | $35.60 \pm 0.56\%$ |
| | ✓ | $\mathbf{66.32 \pm 0.80}\%$ | $\mathbf{47.47 \pm 0.75}\%$ | $31.61 \pm 0.53\%$ | $\mathbf{55.77 \pm 0.79}\%$ | $35.95 \pm 0.58\%$ |

| 5-way 5-Shot | FT | mini-ImageNet | CUB | Cars | Places | Plantae |
|---|---|---|---|---|---|---|
| MatchingNet | - | $70.96 \pm 0.65\%$ | $51.37 \pm 0.77\%$ | $38.99 \pm 0.64\%$ | $63.16 \pm 0.77\%$ | $\mathbf{46.53 \pm 0.68}\%$ |
| | ✓ | $\mathbf{72.53 \pm 0.69}\%$ | $\mathbf{55.23 \pm 0.83}\%$ | $\mathbf{41.24 \pm 0.65}\%$ | $\mathbf{64.55 \pm 0.75}\%$ | $41.69 \pm 0.63\%$ |
| RelationNet | - | $71.00 \pm 0.69\%$ | $57.77 \pm 0.69\%$ | $37.33 \pm 0.68\%$ | $63.32 \pm 0.76\%$ | $44.00 \pm 0.60\%$ |
| | ✓ | $\mathbf{73.78 \pm 0.64}\%$ | $\mathbf{59.46 \pm 0.71}\%$ | $\mathbf{39.91 \pm 0.69}\%$ | $\mathbf{66.28 \pm 0.72}\%$ | $\mathbf{45.08 \pm 0.59}\%$ |
| GNN | - | $80.87 \pm 0.56\%$ | $62.25 \pm 0.65\%$ | $44.28 \pm 0.63\%$ | $70.84 \pm 0.65\%$ | $52.53 \pm 0.59\%$ |
| | ✓ | $\mathbf{81.98 \pm 0.55}\%$ | $\mathbf{66.98 \pm 0.68}\%$ | $\mathbf{44.90 \pm 0.64}\%$ | $\mathbf{73.94 \pm 0.67}\%$ | $\mathbf{53.85 \pm 0.62}\%$ |

metric-based models and is widely adopted in several recent frameworks (Rusu et al., 2019; Gidaris & Komodakis, 2018; Lifchitz et al., 2019).

## 4.2 FEATURE-WISE TRANSFORMATION WITH MANUAL PARAMETER TUNING

We train the model using the mini-ImageNet dataset and evaluate the trained model with four other unseen domains: CUB, Cars, Places, and Plantae. We add the proposed feature-wise transformation layers after the last batch normalization layer of all the residual blocks in the feature encoder $E$ during the training stage. We empirically set $\theta_\gamma$ and $\theta_\beta$ in all feature-wise transformation layers to be $0.3$ and $0.5$, respectively. Table 1 shows the metric-based model trained with the feature-wise transformation layers performs favorably against the individual baselines. We attribute the improvement of generalization to the use of the proposed layers for making the feature encoder $E$ produce more diverse feature distributions in the training stage. As a by-product, we also observe the improvement on the *seen* domain (i.e., mini-ImageNet) since there is still a slight discrepancy between the feature distributions extracted from the training and testing sets of the same domain. It is noteworthy that we also compare the proposed method with several recent approaches (e.g., Lee et al. (2019)) in Table 8 and Table 9. With the proposed feature-wise transformation layers, the GNN (Garcia & Bruna, 2018) model performs favorably against the state-of-the-art frameworks on both the seen domain (i.e., mini-ImageNet) and unseen domains.

## 4.3 GENERALIZATION FROM MULTIPLE DOMAINS

Here we validate the effectiveness of the proposed learning-to-learn algorithm for optimizing the hyper-parameters of the feature-wise transformation layers. We compare the metric-model trained with the proposed learning procedure to the model trained with the *pre-determined* feature-wise transformation layers. The leave-one-out setting is used to select one domain from the CUB, Cars, Places, and Plantae as the unseen domain for the evaluation. The mini-ImageNet and the remaining domains serve as the seen domains for training the model. Since we select the training iteration according to the validation performance on the mini-ImageNet domain for evaluation, we do not consider the mini-ImageNet as the unseen domain. We present the results in Table 2. We denote FT and LFT as applying pre-determined feature-wise transformation layers and those layers optimized with the proposed learning-to-learn algorithm, respectively. The models optimized with proposed learning scheme outperforms those trained with the pre-determined feature-wise transformation layers since the optimized feature-wise transformation layers can better capture the variation of feature distributions across different domains. Table 1 and Table 2 show that the proposed feature-wise transformation layers together with the learning-to-learn algorithm effectively mitigate the domain shift problem for metric-based frameworks.

Table 2: **Few-shot classification results trained with multiple datasets.** We use the leave-one-out setting to select the unseen domain and train the model as well as the feature-wise transformation layers using Algorithm 1. FT and LFT indicate applying the pre-determined and learning-to-learned feature-wise transformation, respectively.

| 5-way 1-Shot | | CUB | Cars | Places | Plantae |
|---|---|---|---|---|---|
| MatchingNet | - | $37.90 \pm 0.55\%$ | $28.96 \pm 0.45\%$ | $49.01 \pm 0.65\%$ | $33.21 \pm 0.51\%$ |
| | FT | $41.74 \pm 0.59\%$ | $28.30 \pm 0.44\%$ | $48.77 \pm 0.65\%$ | $32.15 \pm 0.50\%$ |
| | LFT | $\mathbf{43.29 \pm 0.59}\%$ | $\mathbf{30.62 \pm 0.48}\%$ | $\mathbf{52.51 \pm 0.67}\%$ | $\mathbf{35.12 \pm 0.54}\%$ |
| RelationNet | - | $44.33 \pm 0.59\%$ | $29.53 \pm 0.45\%$ | $47.76 \pm 0.63\%$ | $33.76 \pm 0.52\%$ |
| | FT | $44.67 \pm 0.58\%$ | $30.38 \pm 0.47\%$ | $48.40 \pm 0.64\%$ | $35.40 \pm 0.53\%$ |
| | LFT | $\mathbf{48.38 \pm 0.63}\%$ | $\mathbf{32.21 \pm 0.51}\%$ | $\mathbf{50.74 \pm 0.66}\%$ | $35.00 \pm 0.52\%$ |
| GNN | - | $49.46 \pm 0.73\%$ | $32.95 \pm 0.56\%$ | $51.39 \pm 0.80\%$ | $37.15 \pm 0.60\%$ |
| | FT | $48.24 \pm 0.75\%$ | $33.26 \pm 0.56\%$ | $54.81 \pm 0.81\%$ | $37.54 \pm 0.62\%$ |
| | LFT | $\mathbf{51.51 \pm 0.80}\%$ | $\mathbf{34.12 \pm 0.63}\%$ | $\mathbf{56.31 \pm 0.80}\%$ | $\mathbf{42.09 \pm 0.68}\%$ |
| 5-way 5-Shot | | CUB | Cars | Places | Plantae |
| MatchingNet | - | $51.92 \pm 0.80\%$ | $39.87 \pm 0.51\%$ | $61.82 \pm 0.57\%$ | $47.29 \pm 0.51\%$ |
| | FT | $56.29 \pm 0.80\%$ | $39.58 \pm 0.54\%$ | $62.32 \pm 0.58\%$ | $46.48 \pm 0.52\%$ |
| | LFT | $\mathbf{61.41 \pm 0.57}\%$ | $\mathbf{43.08 \pm 0.55}\%$ | $\mathbf{64.99 \pm 0.59}\%$ | $\mathbf{48.32 \pm 0.57}\%$ |
| RelationNet | - | $62.13 \pm 0.74\%$ | $40.64 \pm 0.54\%$ | $64.34 \pm 0.57\%$ | $46.29 \pm 0.56\%$ |
| | FT | $63.64 \pm 0.77\%$ | $42.24 \pm 0.57\%$ | $65.42 \pm 0.58\%$ | $47.81 \pm 0.51\%$ |
| | LFT | $\mathbf{64.99 \pm 0.54}\%$ | $\mathbf{43.44 \pm 0.59}\%$ | $\mathbf{67.35 \pm 0.54}\%$ | $\mathbf{50.39 \pm 0.52}\%$ |
| GNN | - | $69.26 \pm 0.68\%$ | $48.91 \pm 0.67\%$ | $72.59 \pm 0.67\%$ | $58.36 \pm 0.68\%$ |
| | FT | $70.37 \pm 0.68\%$ | $47.68 \pm 0.63\%$ | $74.48 \pm 0.70\%$ | $57.85 \pm 0.68\%$ |
| | LFT | $\mathbf{73.11 \pm 0.68}\%$ | $\mathbf{49.88 \pm 0.67}\%$ | $\mathbf{77.05 \pm 0.65}\%$ | $\mathbf{58.84 \pm 0.66}\%$ |

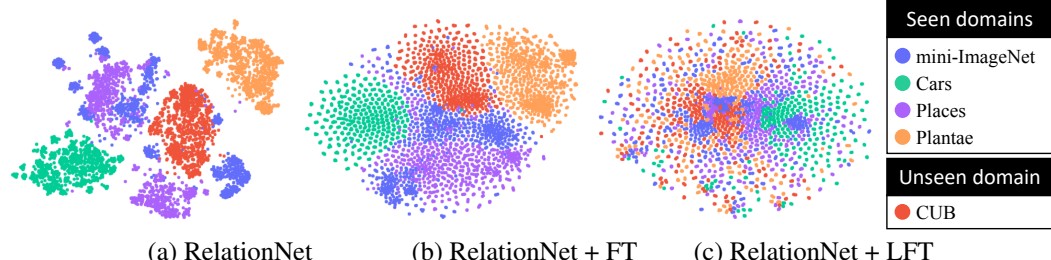

(a) RelationNet      (b) RelationNet + FT      (c) RelationNet + LFT

Figure 3: **T-SNE visualization of the image features extracted from tasks in different domains.** We show the t-SNE visualization of the features extracted by the (a) original feature encoder $E$, (b) feature encoder with pre-determined feature-wise transformation layers, and (c) feature encoder with learning-to-learned feature-wise transformation.

Note since the proposed learning-to-learn approach optimizes the hyper-parameters via stochastic gradient descent, it may not find the global minimum that achieves the best performance. It is certainly possible to manually find another set of hyper-parameter setting that achieves better performance. However, this requires meticulous and computationally expensive hyper-parameter tuning. Specifically, the dimension of the hyper-parameters $\theta_\gamma$ and $\theta_\beta$ is $c_i \times 1 \times 1$ for the $i$-th feature-wise transformation layer in the feature encoder, where $c_i$ is the number of feature channels. As there are $n$ feature-wise transformation layers in the feature encoder $E$, we need to perform the hyper-parameter search in a $(c_1 + c_2 \cdots + c_n) \times 2$-dimensional space. In practice, the dimension of the the search space is 1920.

**Visualizing feature-wise transformed features.** To demonstrate that the proposed feature-wise transformation layers can simulate various feature distributions extracted from the task in different domains, we show the t-SNE visualizations of the image features extracted by the feature encoder in the RelationNet (Sung et al., 2018) model in Figure 3. The model is trained with 5-way 5-shot classification setting on the mini-ImageNet, Cars, Places, and Plantae domains (i.e., corresponding

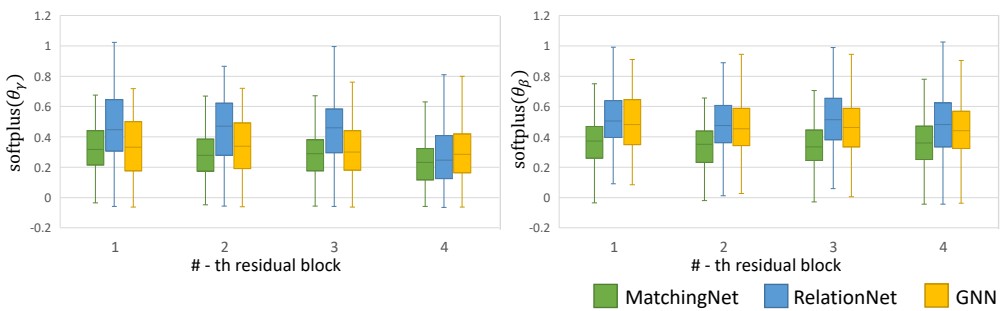

Figure 4: **Visualization of the feature-wise transformation layers.** We show the quartile visualization of the activations $\text{softplus}(\theta_\gamma)$ and $\text{softplus}(\theta_\beta)$ from each feature-wise transformation layer that are optimized by the proposed learning-to-learn algorithm.

to the fifth block of the second column in Table 2). We observe that the distance between features extracted from different domains becomes smaller with the help of feature-wise transformation layers. Furthermore, the proposed learning-to-learn scheme can further help the feature-wise transformation layers capture the variation of feature distributions from various domains, thus close the domain gap and improve the generalization ability of metric-based models.

**Visualizing feature-wise transformation layers.**    To better understand how the learned feature-wise transformation layers operate, we show the values of the $\text{softplus}(\theta_\gamma)$ and $\text{softplus}(\theta_\gamma)$ in the feature-wise transformation layer. Figure 4 presents the visualization. The values of scaling terms $\text{softplus}(\theta_\gamma)$ tend to become smaller in the deeper layers, particularly for those in the last residual block. On the other hand, the depth of the layer does not seem to have an apparent impact on the distributions of the bias terms $\text{softplus}(\theta_\beta)$. The distributions are also different across different metric-based classification methods. These results suggest the importance of the proposed learning-to-learn algorithm because there does not exist a set of optimal hyper-parameters of the feature-wise transformation layers which work well with all metric-based approaches.

## 5    CONCLUSIONS

We propose a method to effectively enhance metric-based few-shot classification frameworks under domain shifts. The core idea of our method lies in using the feature-wise transformation layer to simulate various feature distributions extracted from the tasks in different domains. We develop a learning-to-learn approach for optimizing the hyper-parameters of the feature-wise transformation layers by simulating the generalization process using multiple seen domains. From extensive experiments, we demonstrate that our technique is applicable to different metric-based few-shot classification algorithms and show consistent improvement over the baselines.

## 6    ACKNOWLEDGEMENTS

This work is supported in part by the NSF CAREER Grant #1149783, the NSF Grant #1755785, and gifts from Google.

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

# A  APPENDIX

## A.1  DATASET COLLECTION

We use five few-shot classification datasets in all of our experiments: mini-ImageNet, CUB, Cars, Places, and Plantae. We follow the setting in Ravi & Larochelle (2017) and Hilliard et al. (2018) to process mini-ImageNet and CUB datasets. As for the other datasets, we manually process the dataset by random splitting the classes. The number of training, validation, testing categories for each dataset are summarized in Table 3.

Table 3: **Summarization of the datasets (domains)**. We additionally collect and split the Cars, Places, and Plantae datasets.

| Datasets | mini-ImageNet | CUB | Cars | Places | Plantae |
|---|---|---|---|---|---|
| Source | Deng et al. (2009) | Welinder et al. (2010) | Krause et al. (2013) | Zhou et al. (2017) | Van Horn et al. (2018) |
| # Training categories | 64 | 100 | 98 | 183 | 100 |
| # Validation categories | 16 | 50 | 49 | 91 | 50 |
| # Testing categories | 20 | 50 | 49 | 91 | 50 |
| Split setting | Ravi & Larochelle (2017) | Hilliard et al. (2018) | randomly split | randomly split | randomly split |

## A.2  ADDITIONAL IMPLEMENTATION DETAILS

We use the implementation and adopt the setting of hyper-parameters from Chen et al. (Chen et al., 2019a).[4] We train the metric-based model and feature-wise transformation layers with a learning rate of $0.001$ and $40,000$ iterations. For feature-wise transformation layers, we apply L2 regularization with a weight of $10^{-8}$. The number of inner iterations adopted in the learning-to-learn scheme is set to be 1.

**Matching Network.** We cannot utilize the MatchingNet implementation from Chen et al. (Chen et al., 2019a) since they applied the Pytorch built-in LSTM module, which does not support second-order backpropagation. Without the second-order backpropagation, we are unable to optimize the feature augmentation layers using the proposed learning-to-learn algorithm. As a result, we re-implement the LSTM module for the MatchingNet model. To verify the correctness of our implementation, we evaluate the 5-way 5-shot performance with the ResNet-10 (He et al., 2016) backbone network on the mini-ImageNet dataset (Ravi & Larochelle, 2017). Our implementation reports $68.88 \pm 0.69\%$ accuracy, which is similar to the result (i.e., $68.82 \pm 0.65\%$) reported by Chen et al. (Chen et al., 2019a). We make the source code and datasets public available to foster future progress in this field.[5]

## A.3  ADDITIONAL EXPERIMENTAL RESULTS

**Ablation study on pre-trained metric encoder.** As described in Section 4.1, we pre-trained the metric encoder $E$ by minimizing the cross-entropy classification loss using the $64$ training categories from the mini-ImageNet dataset. To understand the impact of the pre-training, we conduct an ablation study using the leave-one-out experiment illustrated in Section 4.3. As shown in Table 4, pre-training the metric encoder $E$ substantially improve the few-shot classification performance of metric-based frameworks. Note that such a pre-training process is also adopted by several recent frameworks (Rusu et al., 2019; Gidaris & Komodakis, 2018; Lifchitz et al., 2019) to boost the few-shot classification performance.

**Number of ways in testing stage.** In this experiment, we consider a practical scenario that the number of ways $N_w$ in the testing phase is *different* from that in the training stage. Since the GNN (Garcia & Bruna, 2018) framework requires the numbers of ways to be consistent in the training and testing, we evaluate the MatchingNet (Vinyals et al., 2016) and RelationNet (Sung et al., 2018) model with this setting. Table 5 reports the performances of the models trained on the mini-ImageNet, Cars, Places, and Plantae domains under the 5-way 5-shot setting (i.e., corresponding

---

[4] https://github.com/wyharveychen/CloserLookFewShot
[5] https://github.com/hytseng0509/CrossDomainFewShot

Table 4: **Ablation study on pre-trained metric encoder.** We conduct leave-one-out setting to select the unseen domain to study the effectiveness of pre-training the feature encoder $E$ on the mini-ImageNet dataset.

| 1-Shot | Pre-trained | CUB | Cars | Places | Plantae |
|---|---|---|---|---|---|
| MatchingNet | - | $37.37 \pm 0.55\%$ | $\mathbf{30.60 \pm 0.51}\%$ | $41.42 \pm 0.59\%$ | $31.93 \pm 0.51\%$ |
| | ✓ | $37.90 \pm 0.55\%$ | $28.96 \pm 0.45\%$ | $\mathbf{49.01 \pm 0.65}\%$ | $\mathbf{33.21 \pm 0.51}\%$ |
| RelationNet | - | $38.46 \pm 0.56\%$ | $30.77 \pm 0.51\%$ | $37.49 \pm 0.58\%$ | $32.86 \pm 0.53\%$ |
| | ✓ | $\mathbf{44.33 \pm 0.59}\%$ | $29.53 \pm 0.45\%$ | $\mathbf{47.76 \pm 0.63}\%$ | $\mathbf{33.76 \pm 0.52}\%$ |
| GNN | - | $37.21 \pm 0.63\%$ | $29.01 \pm 0.56\%$ | $36.06 \pm 0.62\%$ | $34.99 \pm 0.63\%$ |
| | ✓ | $\mathbf{49.46 \pm 0.73}\%$ | $\mathbf{32.95 \pm 0.56}\%$ | $\mathbf{51.39 \pm 0.80}\%$ | $\mathbf{37.15 \pm 0.60}\%$ |

| 5-Shot | Pre-trained | CUB | Cars | Places | Plantae |
|---|---|---|---|---|---|
| MatchingNet | - | $49.83 \pm 0.55\%$ | $39.41 \pm 0.53\%$ | $59.18 \pm 0.60\%$ | $43.53 \pm 0.53\%$ |
| | ✓ | $\mathbf{51.92 \pm 0.80}\%$ | $39.87 \pm 0.51\%$ | $\mathbf{61.82 \pm 0.57}\%$ | $\mathbf{47.29 \pm 0.51}\%$ |
| RelationNet | - | $55.85 \pm 0.55\%$ | $\mathbf{42.55 \pm 0.58}\%$ | $59.85 \pm 0.54\%$ | $45.24 \pm 0.55\%$ |
| | ✓ | $\mathbf{62.13 \pm 0.74}\%$ | $40.64 \pm 0.54\%$ | $\mathbf{64.34 \pm 0.57}\%$ | $\mathbf{46.29 \pm 0.56}\%$ |
| GNN | - | $60.13 \pm 0.64\%$ | $43.60 \pm 0.67\%$ | $56.67 \pm 0.64\%$ | $49.17 \pm 0.62\%$ |
| | ✓ | $\mathbf{69.26 \pm 0.68}\%$ | $\mathbf{48.91 \pm 0.67}\%$ | $\mathbf{72.59 \pm 0.67}\%$ | $\mathbf{58.36 \pm 0.68}\%$ |

Table 5: **Few-shot classification results under various numbers of ways in testing stage.** We compare the 5-shot performance under various number of ways in the testing phase. The CUB dataset is select as the testing (unseen) domain. All the models are trained with 5-way 5-shot setting.

| 5-Shot | | CUB 2-way | CUB 5-way | CUB 10-way | CUB 20-way |
|---|---|---|---|---|---|
| MatchingNet | - | $78.46 \pm 0.78\%$ | $51.92 \pm 0.80\%$ | $38.22 \pm 0.38\%$ | $26.17 \pm 0.24\%$ |
| | FT | $80.74 \pm 0.77\%$ | $56.29 \pm 0.80\%$ | $41.09 \pm 0.39\%$ | $29.19 \pm 0.24\%$ |
| | LFT | $\mathbf{83.88 \pm 0.72}\%$ | $\mathbf{61.41 \pm 0.57}\%$ | $\mathbf{45.69 \pm 0.39}\%$ | $\mathbf{32.81 \pm 0.23}\%$ |
| RelationNet | - | $84.25 \pm 0.72\%$ | $62.13 \pm 0.74\%$ | $47.15 \pm 0.40\%$ | $34.52 \pm 0.24\%$ |
| | FT | $\mathbf{85.48 \pm 0.69}\%$ | $63.64 \pm 0.77\%$ | $48.35 \pm 0.38\%$ | $35.30 \pm 0.24\%$ |
| | LFT | $85.44 \pm 0.72\%$ | $\mathbf{64.99 \pm 0.54}\%$ | $\mathbf{49.90 \pm 0.40}\%$ | $\mathbf{37.20 \pm 0.25}\%$ |

to the fourth and fifth block of the second column in Table 2). Our proposed learning-to-learned feature-wise transformation layers are capable of improving the generalization of metric-based models to the unseen domain under various numbers of ways in the testing stage.

**Pre-determined hyper-parameters of feature-wise transformation layers.** We demonstrate the difficulty to hand-tune the hyper-parameters of the proposed feature-wise transformation layers in this experiment. Different from the setting described in Section 4.2, we set the hyper-parameters $\theta_\gamma$ and $\theta_\beta$ in all feature-wise transformation layers to be 1. The model is trained under 5-way setting using the mini-ImageNet domain, and evaluate it on the other domains. We report the 1-shot and 5-shot performance in Table 6. We denote applying feature-wise transformation layers with $\{\theta_\gamma, \theta_\beta\} = \{0.3, 0.5\}$ as FT and those with $\{\theta_\gamma, \theta_\beta\} = \{1, 1\}$ as FT*. We observe that the metric-based models applied with FT perform favorably against to those applied with FT*. In several cases, applying FT* even yields inferior results compared to the original training without the feature-wise transformation layers. This suggests the difficulty of hand-tuning the hyper-parameters and the importance of the proposed learning-to-learn scheme for optimizing the hyper-parameters of the feature-wise transformation layers.

**Hyper-parameter initialization for learning-to-learn.** For all the experiments, we initialize the parameters $\theta_\gamma$ and $\theta_\beta$ to 0.3 and 0.5, which we empirically determine in Section 4.2, to train the feature-wise transformation layers. In practice, we find that the cross-domain performance is not sensitive as long as the initialized values are within the same order (e.g., 0.1 and 0.3). Here we report the results of training the RelationNet with the initialization $\{\theta_\gamma, \theta_\beta\} = \{0.1, 0.3\}$. In this experiment, we use the CUB dataset as the unseen domain for evaluation and conduct the training

Table 6: **Few-shot classification results by applying different pre-determined hyper-parameters of feature-wise transformation layers.** We train the model on the mini-ImageNet with a different set of pre-determined hyper-parameters of feature-wise transformation layers. FT and FT* indicate that we apply the feature-wise transformation layers with hyper-parameters $\{\theta_\gamma, \theta_\beta\}$ to be $\{0.3, 0.5\}$ and $\{1, 1\}$, respectively.

| 1-Shot | | mini-ImageNet | CUB | Cars | Places | Plantae |
|---|---|---|---|---|---|---|
| MatchingNet | FT | $58.76 \pm 0.61\%$ | $36.61 \pm 0.53\%$ | $29.82 \pm 0.44\%$ | $51.07 \pm 0.68\%$ | $33.48 \pm 0.50\%$ |
| | FT* | $51.66 \pm 0.64\%$ | $31.74 \pm 0.51\%$ | $27.08 \pm 0.41\%$ | $45.04 \pm 0.64\%$ | $28.73 \pm 0.42\%$ |
| RelationNet | FT | $58.64 \pm 0.85\%$ | $44.07 \pm 0.77\%$ | $28.63 \pm 0.59\%$ | $50.68 \pm 0.87\%$ | $33.14 \pm 0.62\%$ |
| | FT* | $57.45 \pm 0.66\%$ | $40.20 \pm 0.53\%$ | $29.15 \pm 0.45\%$ | $49.40 \pm 0.64\%$ | $33.21 \pm 0.47\%$ |
| GNN | FT | $66.32 \pm 0.80\%$ | $47.47 \pm 0.75\%$ | $31.61 \pm 0.53\%$ | $55.77 \pm 0.79\%$ | $35.95 \pm 0.58\%$ |
| | FT* | $62.63 \pm 0.76\%$ | $44.61 \pm 0.66\%$ | $31.56 \pm 0.52\%$ | $53.39 \pm 0.74\%$ | $36.73 \pm 0.57\%$ |
| **5-Shot** | | mini-ImageNet | CUB | Cars | Places | Plantae |
| MatchingNet | FT | $72.53 \pm 0.69\%$ | $55.23 \pm 0.83\%$ | $41.24 \pm 0.65\%$ | $64.55 \pm 0.75\%$ | $41.69 \pm 0.63\%$ |
| | FT* | $64.93 \pm 0.60\%$ | $42.83 \pm 0.61\%$ | $32.19 \pm 0.48\%$ | $59.47 \pm 0.63\%$ | $39.61 \pm 0.49\%$ |
| RelationNet | FT | $73.78 \pm 0.64\%$ | $59.46 \pm 0.71\%$ | $39.91 \pm 0.69\%$ | $66.28 \pm 0.72\%$ | $45.08 \pm 0.59\%$ |
| | FT* | $72.79 \pm 0.64\%$ | $59.18 \pm 0.57\%$ | $40.54 \pm 0.54\%$ | $65.73 \pm 0.52\%$ | $43.64 \pm 0.49\%$ |
| GNN | FT | $81.98 \pm 0.55\%$ | $66.98 \pm 0.68\%$ | $44.90 \pm 0.64\%$ | $73.94 \pm 0.67\%$ | $53.85 \pm 0.62\%$ |
| | FT* | $82.40 \pm 0.58\%$ | $66.33 \pm 0.73\%$ | $47.63 \pm 0.64\%$ | $75.48 \pm 0.65\%$ | $51.92 \pm 0.59\%$ |

Table 7: **Few-shot classification results by applying the learning-to-learn approach trained with a single seen domain.** We attempt to conduct the proposed learning-to-learn trainin with as singe seen domain, denoted as LFT*. We train the model using the mini-ImageNet dataset and report the 5-way 5-shot classification accuracy.

| 5-way 5-Shot | | mini-ImageNet | CUB | Cars | Places | Plantae |
|---|---|---|---|---|---|---|
| RelationNet | FT | $73.78 \pm 0.64\%$ | $59.46 \pm 0.71\%$ | $39.91 \pm 0.69\%$ | $66.28 \pm 0.72\%$ | $45.08 \pm 0.59\%$ |
| RelationNet | LFT* | $73.50 \pm 0.50\%$ | $58.19 \pm 0.52\%$ | $39.35 \pm 0.54\%$ | $66.17 \pm 0.57\%$ | $46.75 \pm 0.51\%$ |

described in Algorithm 1. The 5-way 5-shot classification accuracy on the CUB dataset is $64.79 \pm 0.55\%$, which is similar to the one we report in Table 2 ( i.e., $64.99 \pm 0.54\%$).

**Learning-to-learn using a single domain.** The proposed learning-to-learn method requires multiple domains for training. Here we apply the learning-to-learn method based on one single domain. More specifically, we randomly sample two different tasks from the mini-ImageNet dataset in each iteration of the training process described in Algorithm 1. One task serves as the pseudo-seen task, while the other one serves as the pseudo-unseen task. We train the RelationNet model with the above-mentioned setting on 5-way 5-shot classification using the mini-ImageNet dataset only. As shown in Table 7, the performance improvement of the RelationNet model is not significant compared to the model trained with pre-determined hyper-parameters $\{\theta_\gamma, \theta_\beta\} = \{0.3, 0.5\}$. This suggests that utilizing one single domain for learning the feature-wise transformation is not as effective as that using multiple domains (demonstrated in Table 2). The reason is that during the training phase, the discrepancy of the feature distributions extracted from the psudo-seen and pseudo-unseen tasks is not as significant since these two tasks are sampled from the same domain. As a result, the hyper-parameters $\{\theta_\gamma, \theta_\beta\}$ cannot be effectively optimized to capture the variation of feature distributions sampled from various domains.

**Comparison with the state-of-the-art few-shot classification on the mini-ImageNet.** We compare the metric-based frameworks applied with the proposed feature-wise transformation layers to the state-of-the-art few-shot classification methods in Table 8. In this experiment, we train the model with the pre-determined hyper-parameters of feature-wise transformation layers on the training set of the mini-ImageNet (Ravi & Larochelle, 2017) dataset. Note that we do not use the *learned* version of the feature-wise transformation layers in the training to ensure fair comparison. Combining Table 4 and Table 8, we observe that the metric-based frameworks train with 1) pre-trained feature encoder, and 2) feature-wise transformation layers with carefully hand-tuned hyper-parameters can demonstrate competitive performance.

Table 8: **Comparison to the state-of-the-art few-shot classification algorithms.** We compare the metric-based frameworks applied with the proposed feature-wise transformation layers using pre-determined hyper-parameter $\{\theta_\gamma, \theta_\beta\} = \{0.3, 0.5\}$ (denoted as FT) to other state-of-the-art few-shot classification methods. Note that all the methods are trained only on the mini-ImageNet dataset. To ensure fair comparisons with other methods, we are unable to use the *learned* version of the feature-wise transformation layers described in Section 3.3. By augmenting existing metric-based few-shot classification models with the proposed feature-wise transformation layer, we obtain competitive performance when compared with many recent and more complicated methods. The best results in each block are highlighted in bold.

| backbone | method | | 5-way 1-shot | 5-way 5-shot |
|---|---|---|---|---|
| ResNet-12 | TADAM (Oreshkin et al., 2018) | | $58.50 \pm 0.30\%$ | $76.70 \pm 0.30\%$ |
| | DC (Lifchitz et al., 2019) | | $62.53 \pm 0.19\%$ | $78.95 \pm 0.13\%$ |
| | DC + IMP (Lifchitz et al., 2019) | | - | $79.77 \pm 0.19\%$ |
| | MetaOptNet-SVM-trainval (Lee et al., 2019) | | $\mathbf{64.09 \pm 0.62}\%$ | $\mathbf{80.00 \pm 0.45}\%$ |
| WRN-28 | Qiao et al. (2018) | | $59.60 \pm 0.41\%$ | $77.74 \pm 0.19\%$ |
| | LEO (Rusu et al., 2019) | | $61.76 \pm 0.08\%$ | $77.59 \pm 0.12\%$ |
| ResNet-10 | MatchingNet | - | $59.10 \pm 0.64\%$ | $70.96 \pm 0.65\%$ |
| | | FT | $58.76 \pm 0.61\%$ | $72.53 \pm 0.69\%$ |
| | RelationNet | - | $57.80 \pm 0.88\%$ | $71.00 \pm 0.69\%$ |
| | | FT | $58.64 \pm 0.85\%$ | $73.78 \pm 0.64\%$ |
| | GNN | - | $60.77 \pm 0.75\%$ | $80.87 \pm 0.56\%$ |
| | | FT | $\mathbf{66.32 \pm 0.80}\%$ | $\mathbf{81.98 \pm 0.55}\%$ |

Table 9: **Evaluation with the state-of-the-art approach under the cross-domain setting.** We evaluate the metric-based frameworks with the proposed feature-wise transformation layers using pre-determined hyper-parameter $\{\theta_\gamma, \theta_\beta\} = \{0.3, 0.5\}$ (denoted as FT) against the state-of-the-art MetaOptNet-SVM-trainval Lee et al. (2019) method. Note that all the methods are trained only on the mini-ImageNet dataset. To ensure fair comparisons with other methods, we do not use the *learned* version of the feature-wise transformation layers described in Section 3.3. By augmenting the existing metric-based few-shot classification models with the proposed feature-wise transformation layer, we obtain competitive performance when compared with recent and more complicated methods. The best results are highlighted in bold.

| method | | CUB | Cars | Places | Plantae |
|---|---|---|---|---|---|
| MetaOptNet-SVM-trainval | | $54.67 \pm 0.56\%$ | $\mathbf{45.90 \pm 0.49}\%$ | $65.83 \pm 0.57\%$ | $46.48 \pm 0.52\%$ |
| MatchingNet | - | $51.37 \pm 0.77\%$ | $38.99 \pm 0.64\%$ | $63.16 \pm 0.77\%$ | $46.53 \pm 0.68\%$ |
| | FT | $55.23 \pm 0.83\%$ | $41.24 \pm 0.65\%$ | $64.55 \pm 0.75\%$ | $41.69 \pm 0.63\%$ |
| RelationNet | - | $57.77 \pm 0.69\%$ | $37.33 \pm 0.68\%$ | $63.32 \pm 0.76\%$ | $44.00 \pm 0.60\%$ |
| | FT | $59.46 \pm 0.71\%$ | $39.91 \pm 0.69\%$ | $66.28 \pm 0.72\%$ | $45.08 \pm 0.59\%$ |
| GNN | - | $62.25 \pm 0.65\%$ | $44.28 \pm 0.63\%$ | $70.84 \pm 0.65\%$ | $52.53 \pm 0.59\%$ |
| | FT | $\mathbf{66.98 \pm 0.68}\%$ | $44.90 \pm 0.64\%$ | $\mathbf{73.94 \pm 0.67}\%$ | $\mathbf{53.85 \pm 0.62}\%$ |

**Comparison to the state-of-the-art few-shot classification under domain shift.** We evaluate the metric-based frameworks with the proposed feature-wise transformation layers and the state-of-the-art MetaOptNet approach Lee et al. (2019). We use the model trained on the mini-ImageNet dataset provided by the authors for the evaluation on the other datasets.[6] As shown in Table 9, while the MetaOptNet method achieves state-of-the-art performance on the mini-ImageNet dataset, this approach also suffers from the domain shifts in the cross-domain setting. Training the GNN framework with the pre-trained feature encoder and the proposed feature-wise transformation layers performs favorably against the MetaOptNet method under the cross-domain setting.

---

[6] https://github.com/kjunelee/MetaOptNet

