# OpenReview forum: "Cross-Domain Few-Shot Classification via Learned Feature-Wise Transformation"
_ICLR.cc/2020/Conference — Accept (Spotlight)_

### Official Review · AnonReviewer2 · 2019-10-21
**Official Blind Review #2**

**Rating:** 6

**Review:**

This paper proposes a feature-wise transformation layer to augment the image features, which is a new regularization of neural networks and leads to better generalization ability of the features. And the proposed method performs well in the few-shot classification problem. Furthermore, the paper develops a learning-to-learn model in the cross-domain setting to choose optimal hyper-parameters of the feature-wise transformation layers. Which leads to consistent improvement in the cross-domain leave-one-out setting.

Although I vote weak accept for this paper, I still have several concerns:
* In equation $(7)$, how to calculate the gradient of $L^{pu}$ w.r.t $\theta_f^t$ in the condition that the $L^{pu}$ is calculated after removing the feature-wise transformation layers from the model? In my understanding, The $L^{pu}$ is not related to $\theta_f^t$ after the removal of the feature-wise transformation layers.
* In Section 4.2, why not choose the Prototypical networks? According to the results in (Chen et al., 2019a), it performs much better in the mini-ImageNet to CUB setting.
* In Section 4.3, what's the initial value of $\theta_{\gamma}$ and $\theta_{\beta}$? Is the initial value sensitive to the performance?
* In Section 4.2, actually, you can learn the $\theta_{\gamma}$ and $\theta_{\beta}$ automatically even in a single domain. For example, use a different batch to update $\theta_{\gamma}$ and $\theta_{\beta}$. What's the performance under this setting?
* There is no direct comparison to the state-of-the-art methods.

Overall, the paper is well written and the figures are well illustrated. The experiments show the effectiveness of the proposed feature-wise transformation layers and the learning-to-learning approach. But the above concerns should be addressed.

**Experience Assessment:**

I have published in this field for several years.

**Review Assessment: Checking Correctness Of Derivations And Theory:**

I carefully checked the derivations and theory.

**Review Assessment: Checking Correctness Of Experiments:**

I carefully checked the experiments.

**Review Assessment: Thoroughness In Paper Reading:**

I read the paper thoroughly.

---

> ### Author Response · Authors · 2019-11-15
> **Responses to AnonReviewer2 -- part 1**
>
> Thanks for your valuable comments. Our responses are as follows:
> ——---
> >>> Comments: In equation (7), how to calculate the gradient of L^{pu} w.r.t \theta^t_f in the condition that the L^{pu} is calculated after removing the feature-wise transformation layers from the model? In my understanding, The L^{pu} is not related to after the removal of the feature-wise transformation layers.
>
> > Response: As described in Equation (5), we first update the parameters in the metric-based model with the feature-wise transformation layers. In other words, the updated model used to calculate the loss L^{pu} in Equation (6) is the function of the feature-wise transformation layers. Therefore, we can calculate the gradient of L^{pu} with respect to the hyper-parameter \theta^t_f in the feature-wise transformation layers by combining Equation (5) and (6).
>
>
> ——---
> >>> Comments: In Section 4.2, why not choose the Prototypical networks? According to the results in (Chen et al., 2019a), it performs much better in the mini-ImageNet to CUB setting.
>
> > Response: As shown in Figure 1, the proposed feature-wise transformation layer aims to improve the generalization ability of the parametric (trainable) metric function M. Therefore, we do not see clear performance improvement based on Prototypical networks since the prototypical networks utilize the euclidean distance as its metric function which is non-trainable. We report the cross-domain performance of the Prototypical networks in the below table.
>
> 5-way 5-shot performance of the Prototypical networks trained on the mini-ImageNet dataset (top row: Prototypical networks, bottom row: Prototypical networks trained with the pre-determined feature-wise transformation layers)
>    mini-ImageNet              CUB                      Cars                     Places                  Plantae
> |  78.05 ± 0.46%  ||  64.45 ± 0.75%  |  47.98 ± 0.73%  |  72.60 ± 0.71%  |  55.19 ± 0.72%  |
> |  78.20 ± 0.63%  ||  64.20 ± 0.58%  |  47.78 ± 0.58%  |  71.21 ± 0.56%  |  54.97 ± 0.54%  |
>
> ——---
> >>> Comments:  In Section 4.3, what's the initial value of \theta_\gamma and \theta_\beta? Is the initial value sensitive to the performance?
>
> > Response: For all the experiments, we initialize the parameters {\theta_\gamma, \theta_\beta} to {0.3, 0.5}, which we empirically determine in Section 4.2, to train the feature-wise transformation layers. In practice, we find that the cross-domain performance is not sensitive as long as the initialized values are within the same order (e.g., {0.1, 0.3}). Here we report the results of training the RelationNet with the initialization {\theta_\beta} = {0.1, 0.3}. In this experiment, we select the CUB dataset as the unseen domain for the evaluation and conduct the training described in Algorithm 1.
>
> {\theta_\beta}          {0.3, 0.5}              {0.1, 0.3}
> RelationNet      | 64.99 +- 0.54% | 64.79 +- 0.55% |
>
>
> ——---
> >>> Comments: In Section 4.2, actually, you can learn the \theta_\gamma and \theta_\beta automatically even in a single domain. For example, use a different batch to update \theta_\gamma and \theta_beta. What's the performance under this setting?
>
> > Response:  As suggested, we sample two different tasks from the mini-ImageNet dataset to conduct the training described in Algorithm 1. More specifically, we randomly sample two tasks in each training iteration. One task serves as the pseudo-seen task, while the other one serves as the pseudo-unseen task. We train the RelationNet model with the above algorithm on 5-way 5-shot classification using the mini-ImageNet dataset only. We show the results as follows.
>
> 5-way 5-shot performance of the RelationNet model trained on the mini-ImageNet dataset (top row: trained with the pre-determined feature-wise transformation layers, bottom row: trained with the learning-to-learned feature-wise transformation layers)
>    mini-ImageNet             CUB                    Cars                    Places                 Plantae
> |  73.78 ± 0.64%  ||  59.46 ± 0.71%  |  39.91 ± 0.69%  |  66.28 ± 0.72%  |  45.08 ± 0.59%  |
> |  73.78 ± 0.50%  ||  58.19 ± 0.52%  |  39.35 ± 0.54%  |  66.17 ± 0.57%  |  46.75 ± 0.51%  |
>
> The improvement on the performance of the RelationNet model is not significant comparing to the model trained with pre-determined hyper-parameters {\theta_\gamma, \theta_\beta} = {0.3, 0.5}. This suggests that utilizing a single seen domain for learning the feature-wise transformation is not as effective as using multiple seen domains (demonstrated in Table 3). The reason is that during the training phase, the discrepancy of the feature distributions extracted from the pseudo-seen and pseudo-unseen tasks is not as significant since these two tasks are sampled from the same domain. As a result, the hyper-parameter {\theta_\gamma, \theta_\beta} cannot be effectively optimized to capture the variation of feature distributions sampled from various domains.

---

> ### Author Response · Authors · 2019-11-15
> **Responses to AnonReviewer2 -- part 2**
>
>
> ——---
> >>> Comments: There is no direct comparison to the state-of-the-art methods.
>
> > Response: We include the results of several state-of-the-art methods [Qiao et al., 2018; Oreshkin et al., 2018; Lifchitz et al., 2019; Lee et al., 2019; Rusu et al., 2019] on the mini-ImageNet dataset in Table 7 in the appendix. Our conclusion is that training the metric-based framework with 1) pre-trained feature encoder, and 2) the proposed feature-wise transformation layers can demonstrate competitive performance. For instance, the GNN approach trained with our pre-determined feature-wise transformation layers shows 66% 1-shot and 81% 5-shot classification accuracies on the mini-ImageNet dataset, which is comparable to the state-of-the-art MetaOptNet [Lee et al., 2019] (with the accuracy of 64% 1-shot and 80% 5-shot).
>
> For the cross-domain setting, we report the performance of MetaOptNet [Lee et al., 2019]. We use the model trained on the mini-ImageNet dataset provided by the authors (https://github.com/kjunelee/MetaOptNet) for the evaluation on other datasets. As shown in the table below, while MetaOptNet [Lee et al., 2019]  achieves state-of-the-art performance on the mini-ImageNet dataset, this approach also suffers from the domain shifts in the cross-domain setting. Training the GNN framework with pre-trained feature encoder and feature-wise transformation layers performs favorably against the MetaOptNet method  [Lee et al., 2019] under the cross-domain setting.
>
> 5-way 5-shot performance of models trained on the mini-ImageNet dataset:
>                                           CUB                       Cars                     Places                  Plantae
> [Lee et al., 2019] |  54.67 ± 0.56%  |  45.90 ± 0.49%  |  65.83 ± 0.57%  |  46.48 ± 0.52%  |
> GNN                      |  62.25 ± 0.65%  |  44.28 ± 0.63%  |  70.84 ± 0.65%  |  52.53 ± 0.59%  |
> GNN + FT             |  66.98 ± 0.68%  |  44.90 ± 0.64%  |  73.94 ± 0.67%  |  53.85 ± 0.62%  |
>
> S. Qiao et al., Few-Shot Image Recognition by Predicting Parameters From Activations, CVPR 2018.
>
> B. Oreshkin et al., TADAM: Task Dependent Adaptive Metric for Improved Few-Shot Learning, NeurIPS 2018.
>
> Y. Lifchitz et al., Dense Classification and Implanting for Few-Shot Learning, CVPR 2019.
>
> K. Lee et al., Meta-Learning with Differentiable Convex Optimization, CVPR 2019.
>
> A. Rusu et al., Meta-Learning with Latent Embedding Optimization, ICLR 2019.

---

### Official Review · AnonReviewer1 · 2019-10-28
**Official Blind Review #1**

**Rating:** 6

**Review:**

In this paper, the authors propose a feature-wise transformation layer for the cross-domain few-shot classification task. Besides, they apply the learning-to-learn procedure to tune the hyperparameters automatically. The primary motivation behind this method does make a lot of sense for me. The reduction of significant shifts in the feature norm could be crucial for successfully transferring the source domain model to the target domain. Also, such a method is method-agnostic, which makes the proposed layer more accessible for all kinds of metric-based query&support pipeline. The extensive experiments for three baselines, MatchingNet, RelationNet, and GNN, indicate the generalizable effectiveness of the proposed layer. Specifically, the visualization and analysis of learned feature space and layer parameters look interesting to me. Remember that the proposed method is designed to minimize the large discrepancy of the feature distribution. Such a study could be beneficial for the reader to understand this paper in depth.

Meanwhile, I have some questions and suggestions for the authors:
1. I am suggesting the authors include more recent state-of-the-art as baselines and comparisons, which could make such submission much stronger.
2. This paper delivers an extensive study of the classification problem, how about the other tasks, which heavily rely on classification head, like detection or segmentation. I think it could be more attractive if the author could also show some improvement for these tasks in either a qualitative or quantitive way.
3. This paper mainly focuses on metric-based few-shot frameworks. At the same time, there are also other two groups, recurrent-based and optimization-based schemes. It could be much better if the authors could also mention the potential of the proposed layer.
4. This paper leverages the learning-to-learn mechanism to tune the hyper-parameters. How about considering the adaptive or dynamic approach. In other words, build the connection between the theta parameter and image feature.

**Experience Assessment:**

I have read many papers in this area.

**Review Assessment: Checking Correctness Of Derivations And Theory:**

I assessed the sensibility of the derivations and theory.

**Review Assessment: Checking Correctness Of Experiments:**

I assessed the sensibility of the experiments.

**Review Assessment: Thoroughness In Paper Reading:**

I read the paper at least twice and used my best judgement in assessing the paper.

---

> ### Author Response · Authors · 2019-11-15
> **Responses to AnonReviewer1 -- part 1**
>
> Thanks for your valuable suggestions. Our responses are as follows:
> ——---
> >>> Comments: I am suggesting the authors include more recent state-of-the-art as baselines and comparisons, which could make such submission much stronger.
>
> > Response: We include the results of several state-of-the-art methods [Qiao et al., 2018; Oreshkin et al., 2018; Lifchitz et al., 2019; Lee et al., 2019; Rusu et al., 2019] on the mini-ImageNet dataset in Table 7 in the appendix. Our conclusion is that training the metric-based framework with 1) pre-trained feature encoder, and 2) the proposed feature-wise transformation layers can demonstrate competitive performance. For instance, the GNN approach trained with our pre-determined feature-wise transformation layers shows 66% 1-shot and 81% 5-shot classification accuracies on the mini-ImageNet dataset, which is comparable to the state-of-the-art MetaOptNet [Lee et al., 2019] (with the accuracy of 64% 1-shot and 80% 5-shot).
>
> For the cross-domain setting, we report the performance of MetaOptNet [Lee et al., 2019]. We use the model trained on the mini-ImageNet dataset provided by the authors (https://github.com/kjunelee/MetaOptNet) for the evaluation on other datasets. As shown in the table below, while MetaOptNet [Lee et al., 2019]  achieves state-of-the-art performance on the mini-ImageNet dataset, this approach also suffers from the domain shifts in the cross-domain setting. Training the GNN framework with pre-trained feature encoder and feature-wise transformation layers performs favorably against the MetaOptNet method  [Lee et al., 2019] under the cross-domain setting.
>
> 5-way 5-shot performance of models trained on the mini-ImageNet dataset:
>                                           CUB                       Cars                     Places                  Plantae
> [Lee et al., 2019] |  54.67 ± 0.56%  |  45.90 ± 0.49%  |  65.83 ± 0.57%  |  46.48 ± 0.52%  |
> GNN                      |  62.25 ± 0.65%  |  44.28 ± 0.63%  |  70.84 ± 0.65%  |  52.53 ± 0.59%  |
> GNN + FT             |  66.98 ± 0.68%  |  44.90 ± 0.64%  |  73.94 ± 0.67%  |  53.85 ± 0.62%  |
>
> S. Qiao et al., Few-Shot Image Recognition by Predicting Parameters From Activations, CVPR 2018.
>
> B. Oreshkin et al., TADAM: Task Dependent Adaptive Metric for Improved Few-Shot Learning, NeurIPS 2018.
>
> Y. Lifchitz et al., Dense Classification and Implanting for Few-Shot Learning, CVPR 2019.
>
> K. Lee et al., Meta-Learning with Differentiable Convex Optimization, CVPR 2019.
>
> A. Rusu et al., Meta-Learning with Latent Embedding Optimization, ICLR 2019.
>
> ——---
> >>> Comments: This paper delivers an extensive study of the classification problem, how about the other tasks, which heavily rely on classification head, like detection or segmentation. I think it could be more attractive if the author could also show some improvement for these tasks in either a qualitative or quantitative way.
>
> > Response:  Take an ICLR submission Meta-RCNN [Anonymous, 2020] as an example, we can construct more datasets by sampling various subsets of the ImageNet-LOC dataset, then investigate the cross-domain performance of Meta-RCNN. Since the DCNN network serves as the feature encoder E, we can apply the proposed feature-wise transformation layers to the DCNN network to produce more diverse feature distributions. Here, the goal is to improve the generalization ability of the attention module and the other modules (RPN, classifier, regressor in the last stage) followed by the DCNN network.
>
> There are also several other recent methods on few-shot semantic segmentation [Rakelly
> et al. 2018] and instance segmentation [Gupta et al. 2019]. We believe that integrating our approach into these models to further advance the performance is an interesting future work.
>
> Anonymous authors, Meta-RCNN: Meta-Learning for Few-Shot Object Detection, ICLR submission 2020.
>
> K. Rakelly et al.: Few-Shot Segmentation Propagation with Guided Networks, preprint 2018
>
> Gupta et al.: LVIS: A Dataset for Large Vocabulary Instance Segmentation, CVPR 2019

---

> ### Author Response · Authors · 2019-11-15
> **Responses to AnonReviewer1 -- part 2**
>
>
> ——---
> >>> Comments: This paper mainly focuses on metric-based few-shot frameworks. At the same time, there are also other two groups, recurrent-based and optimization-based schemes. It could be much better if the authors could also mention the potential of the proposed layer.
>
> > Response:  Since the network architecture of recurrent-based frameworks is different from metric-based approaches, it may not be trivial to apply the proposed feature-wise transformation layers to recurrent-based approaches. However, it is possible to integrate the proposed layers into the feature encoder of optimization-based schemes to improve the generalization ability. Although we have not validated this method yet, there is a concurrent work [Anonymous, 2020] that shows the feasibility of perturbing the features for improving the generalization ability of optimization-based schemes.
>
> Anonymous authors, Meta Dropout: Learning to Perturb Latent Features for Generalization, ICLR submission 2020.
>
> ——---
> >>> Comments: This paper leverages the learning-to-learn mechanism to tune the hyper-parameters. How about considering the adaptive or dynamic approach. In other words, build the connection between the theta parameter and image feature.
>
> > Response:  There have been several recent studies that directly infer the model parameters conditioned on the input features [Qi et al., 2018; Rusu et al., 2019]. Such an image feature-aware approach can potentially further improve the performance. This is certainly a promising future research direction.
>
> Hang Qi et al., Low-Shot Learning with Imprinted Weights, CVPR 2018
>
> Andre A. Rusu et al., Meta-Learning with Latent Embedding Optimization, ICLR 2019

---

### Official Review · AnonReviewer3 · 2019-10-31
**Official Blind Review #3**

**Rating:** 6

**Review:**

The authors argue that existing metric learning approaches, for few-shot learning, may cause overfitting to the feature distributions encoded only from the seen domains and thus fail to generalize to unseen domains. This is a domain generaliation task under a few-shot learning setting. The authors proposed a so-called feature-wise transformation layer and integrate it into some existing metric-based few-shot learning methods, which helps learn more diverse feature distributions for better generalization ability.

I think this is a pioneer work on few-shot learning for domain generalization, which is quite interesting.

In terms of experiments, good performance has been shown by using three existing metric-based few-shot learning methods on a number of settings using five image datasets.

Below list my major concerns:
1. The authors mentioned that it is intuitive to have feature-wise transformation layers produce more diverse feature distributions, which would lead to better generaliztion ability for domain generalization. However, it is not obvious to me. Why the generalization ability would be improved with more diverse features?

2. In the experiments, feature-wise transformation layers with hyper-parameters empirically (\theta_\gamma, \theta_\beta) set as (0.3, 0.5) are compared against learning-to-learn version of the proposed method. Why (0.3, 0.5) was chosen? Is there any chance that another set of (\theta_\gamma, \theta_\beta) may lead to better performance than the learning-to-learn version? It would be much clearer if the authors can provide detailed parameter analysis (e.g. grid search).

3. Algorithm 1 should be quite expensive in time. It is not clear how the algorithm will be terminated. And in general, how many iterations in the algorithm will be until it achieves some convergence?

**Experience Assessment:**

I have published one or two papers in this area.

**Review Assessment: Checking Correctness Of Derivations And Theory:**

I assessed the sensibility of the derivations and theory.

**Review Assessment: Checking Correctness Of Experiments:**

I carefully checked the experiments.

**Review Assessment: Thoroughness In Paper Reading:**

I read the paper thoroughly.

---

> ### Author Response · Authors · 2019-11-15
> **Responses to AnonReviewer3**
>
> Thanks for your valuable comments. Our responses are as follows:
> ——---
> >>> Comments: The authors mentioned that it is intuitive to have feature-wise transformation layers produce more diverse feature distributions, which would lead to better generalization ability for domain generalization. However, it is not obvious to me. Why the generalization ability would be improved with more diverse features?
>
> > Response: As shown in Figure 3(a), since the feature distributions across various domains are significantly different, the metric function M in Figure 1(c), which takes features as input, fail to generalize to unseen domains. To address the problem, we diversify the feature distributions from the seen domains via feature-wise transformation to improve the generalization ability of the metric function M. However, simply adding noise sampled from pre-determined noise distributions to diversify the feature distributions may not be effective. The reason is that the variation of the feature distributions across domains is complex. Therefore, we propose the learning-to-learn algorithm that trains the noise distributions to capture the variation of feature distributions across domains.
>
> ——---
> >>> Comments: In the experiments, feature-wise transformation layers with hyper-parameters empirically (\theta_\gamma, \theta_\beta) set as (0.3, 0.5) are compared against the learning-to-learn version of the proposed method. Why (0.3, 0.5) was chosen?
>
> > Response: We determine the hyper-parameters (\theta_\gamma, \theta_\beta) to be (0.3, 0.5) based on the validation accuracies obtained in the experiments of Table 2. Specifically, we randomly sample five sets of values from the interval of [0.1, 1] (with an interval of 0.1), for both \theta_\gamma and \theta_\beta. We choose to use (0.3, 0.5) according to the validation performance on the mini-ImageNet dataset.
>
> >>> Comments: Is there any chance that another set of (\theta_\gamma, \theta_\beta) may lead to better performance than the learning-to-learn version?
>
> > Response: Since the proposed learning-to-learn approach optimizes the hyper-parameters via stochastic gradient descent, it may not find the global minimum that achieves the best performance. It is certainly possible to manually find another set of hyper-parameter setting that achieves better performance. However, this requires meticulous and computationally expensive hyper-parameter tuning. Specifically, the dimension of the hyper-parameters \theta_\gamma and \theta_\beta is c_i x 1 x1 for the i-th feature-wise transformation layer in the feature encoder, where c_i is the number of feature channels. As there are n feature-wise transformation layers in the encoder, we have to perform the hyper-parameter search in a (c_1 + c_2 ,,, + c_n)*2-dimensional space, which is a 1920-dimensional space in practice.
>
> >>> Comments: It would be much clearer if the authors can provide detailed parameter analysis (e.g. grid-search).
>
> > Response: We agree with the reviewer that a grid-search would indeed provide a clearer picture. Unfortunately, as there are 2 (1, 5-shot) X 4 (unseen domains) X 3 (methods) different settings for the experiments presented in Table 3, we were unable to conduct such a large-scale grid-search analysis for each setting with the limited computational resources at this time. We will provide the detailed grid-search results when preparing the final version of this paper.
>
> ——---
> >>> Comments: Algorithm 1 should be quite expensive in time. It is not clear how the algorithm will be terminated. And in general, how many iterations in the algorithm will be until it achieves some convergence?
>
> > Response: Following the training protocols for all the baseline methods, we terminate the training process outlined in Algorithm 1 at 40,000 iterations. It takes around 3 days on the desktop machine equipped with a single Titan XP GPU, i7 CPU cores, and 16G RAM.

---

### Decision · Program_Chairs · 2019-12-19

**Decision:**

Accept (Spotlight)

**Comment:**

This submission addresses the problem of few-shot classification. The proposed solution centers around metric-based models with a core argument that prior work may lead to learned embeddings which are overfit to the few labeled examples available during learning. Thus, when measuring cross-domain performance, the specialization of the original classifier to the initial domain will be apparent through degraded test time (new domain) performance. The authors therefore, study the problem of domain generalization in the few-shot learning scenario. The main algorithmic contribution is the introduction of a feature-wise transformation layer.

All reviewers suggest to accept this paper. Reviewer 3 says this problem statement is especially novel. Reviewer 1 and 2 had concerns over lack of comparisons with recent state-of-the-art methods. The authors responded with some additional results during the rebuttal phase, which should be included in the final draft.

Overall the AC recommends acceptance, based on the positive comments and the fact that this paper addresses a sufficiently new problem statement.